# Efficient SARS-CoV-2 Surveillance during the Pandemic-Endemic Transition Using PCR-Based Genotyping Assays

Lianne Koets,[a] Karin van Leeuwen,[b] Maaike Derlagen,[c] Jalenka van Wijk,[c] Nadia Keijzer,[c] Jelena D. M. Feenstra,[d] Manoj Gandhi,[d] Oceane Sorel,[d] Thijs J. W. van de Laar,[e,f] Marco H. G. M. Koppelman[a]

aSanquin Research and Lab Services, National Screening Laboratory of Sanquin, Amsterdam, The Netherlands
bSanquin Diagnostics, Department of Phagocytes Diagnostics, Amsterdam, The Netherlands
cSanquin Diagnostics, Department of Immune Cytology, Amsterdam, The Netherlands
dThermo Fisher Scientific, South San Francisco, California, USA
eSanquin Research, Department of Blood-Borne Infections, Amsterdam, The Netherlands
fOnze Lieve Vrouwe Gasthuis, Laboratory of Medical Microbiology, Amsterdam, The Netherlands

**ABSTRACT** Severe acute respiratory syndrome coronavirus-2 (SARS-CoV-2) variants of concern (VOC) pose an increased risk to public health due to higher transmissibility and/or immune escape. In this study, we assessed the performance of a custom TaqMan SARS-CoV-2 mutation panel consisting of 10 selected real-time PCR (RT-PCR) genotyping assays compared to whole-genome sequencing (WGS) for identification of 5 VOC circulating in The Netherlands. SARS-CoV-2 positive samples (N = 664), collected during routine PCR screening ($15 \leq C_T \leq 32$) between May-July 2021 and December 2021-January 2022, were selected and analyzed using the RT-PCR genotyping assays. VOC lineage was determined based on the detected mutation profile. In parallel, all samples underwent WGS with the Ion AmpliSeq SARS-CoV-2 research panel. Among 664 SARS-CoV-2 positive samples, the RT-PCR genotyping assays classified 31.2% as Alpha (N = 207); 48.9% as Delta (N = 325); 19.4% as Omicron (N = 129), 0.3% as Beta (N = 2), and 1 sample as a non-VOC. Matching results were obtained using WGS in 100% of the samples. RT-PCR genotyping assays enable accurate detection of SARS-CoV-2 VOC. Furthermore, they are easily implementable, and the costs and turnaround time are significantly reduced compared to WGS. For this reason, a higher proportion of SARS-CoV-2 positive cases in the VOC surveillance testing can be included, while reserving valuable WGS resources for identification of new variants. Therefore, RT-PCR genotyping assays would be a powerful method to include in SARS-CoV-2 surveillance testing.

**IMPORTANCE** The severe acute respiratory syndrome coronavirus-2 (SARS-CoV-2) genome changes constantly. It is estimated that there are thousands of variants of SARS-CoV-2 by now. Some of those variants, variants of concern (VOC), pose an increased risk to public health due to higher transmissibility and/or immune escape. Pathogen surveillance helps researchers, epidemiologists, and public health officials to monitor the evolution of infectious diseases agents, alert on the spread of pathogens, and develop counter measures like vaccines. The technique used for the pathogen surveillance is called sequence analysis which makes it possible to examine the building blocks of SARS-CoV-2. In this study, a new PCR method based on the detection of specific changes of those building blocks is presented. This method enables a fast, accurate and cheap determination of different SARS-CoV-2 VOC. Therefore, it would be a powerful method to include in SARS-CoV-2 surveillance testing.

**KEYWORDS** RT-PCR genotyping assays, SARS-CoV-2, variant of concern

Address correspondence to Lianne Koets, l.koets@sanquin.nl.

The authors declare no conflict of interest.

Severe acute respiratory syndrome coronavirus-2 (SARS-CoV-2) emerged in Wuhan, China in December 2019 and caused an ongoing pandemic of Coronavirus disease 2019 (COVID-19) (1). Genomic surveillance efforts have led to the identification of many SARS-CoV-2 variants, some of which have been designated by the WHO as variants of concern (VOC) or variants of interest (VOI) based on their characteristics and their potential impact on public health (2).

The SARS-CoV-2 Alpha (B.1.1.7) variant, which originated in the United Kingdom in September 2020, was designated VOC based on increased transmissibility and was the first VOC with a worldwide distribution (3, 4). The Alpha variant carries the characteristic N501Y mutation in the *S* gene encoding the Spike protein, which leads to its increased ACE2 receptor binding affinity and thus higher transmissibility as well as significant resistance to neutralizing antibodies (5, 6). As the pandemic continued and population immunity against SARS-CoV-2 increased through vaccination or infection, 2 other VOC emerged. These variants named Beta (B.1.315) and Gamma (P.1) were first detected in South Africa and Brazil, respectively. Both variants carry the N501Y mutation that was also present in the Alpha variant, as well as additional mutations conferring immune escape including E484K and K417N/T (7–11). In March 2021, a new VOC called Delta (B.1.617.2) was first detected in India. This variant showed increased transmissibility and rapidly became the dominant variant worldwide. Initial studies suggested that Delta variant caused more severe disease in unvaccinated individuals and higher rates of breakthrough infections in those vaccinated against COVID-19 (12, 13). However, a later study showed that the higher number of hospitalized patients during the Delta period was attributable to lower vaccination rates in certain age groups with no significant increase of severe disease outcomes (14). In November 2021, the WHO identified a fifth VOC called Omicron (B.1.1.529) first detected in Southern Africa and was designated VOC status due to high number of mutations in the Spike protein in particular its receptor binding domain (15). Omicron is responsible for the latest wave of COVID-19 cases in most countries reporting record numbers of infections since the start of the pandemic, with recent studies confirming lower disease severity caused by this variant (16).

The presence of overlapping mutations of concern (MOC) in different VOCs was used to develop the described real-time PCR (RT-PCR)-based approach for variant surveillance. As each VOC carries MOCs in different combinations along with additional mutations, genotyping of a few specific genomic positions enables rapid detection and discrimination between different VOCs. While whole-genome sequencing (WGS) is the reference standard for identification of new viral variants, it is resource-intensive, not easily scalable, requires expensive equipment that is not available in most labs, and time-to-result can delay a timely public health response.

In this study, we evaluated the accuracy and sensitivity of a custom TaqMan SARS-CoV-2 mutation panel consisting of 10 RT-PCR genotyping assays classifying MOCs by comparing the results to WGS for SARS-CoV-2 VOC detection in COVID-19 patients. The study was initiated between May and July 2021 when Delta replaced Alpha as the dominant variant in The Netherlands. Subsequently, the detection of MOCs was also evaluated during December 2021 and January 2022, to monitor the emergence of Omicron in The Netherlands.

## RESULTS

**Validation of the RT-PCR genotyping assays using reference material.** To validate the accuracy of selected RT-PCR genotyping assays of the TaqMan SARS-CoV-2 mutation panel, we tested a panel of reference samples (N = 13) provided by The National Institute for Public Health and Environment (RIVM) containing SARS-CoV-2 positive samples with known whole-genome sequences and therefore known types of selected SARS-CoV-2 variants: Alpha, Beta, Gamma, Delta, Epsilon, Eta, Iota, Omicron, and Zeta, as well as lineages B.1.258.21 and B.1.177. The samples were tested using 10 RT-PCR genotyping assays and the results showed a 100% concordance (Table S3).

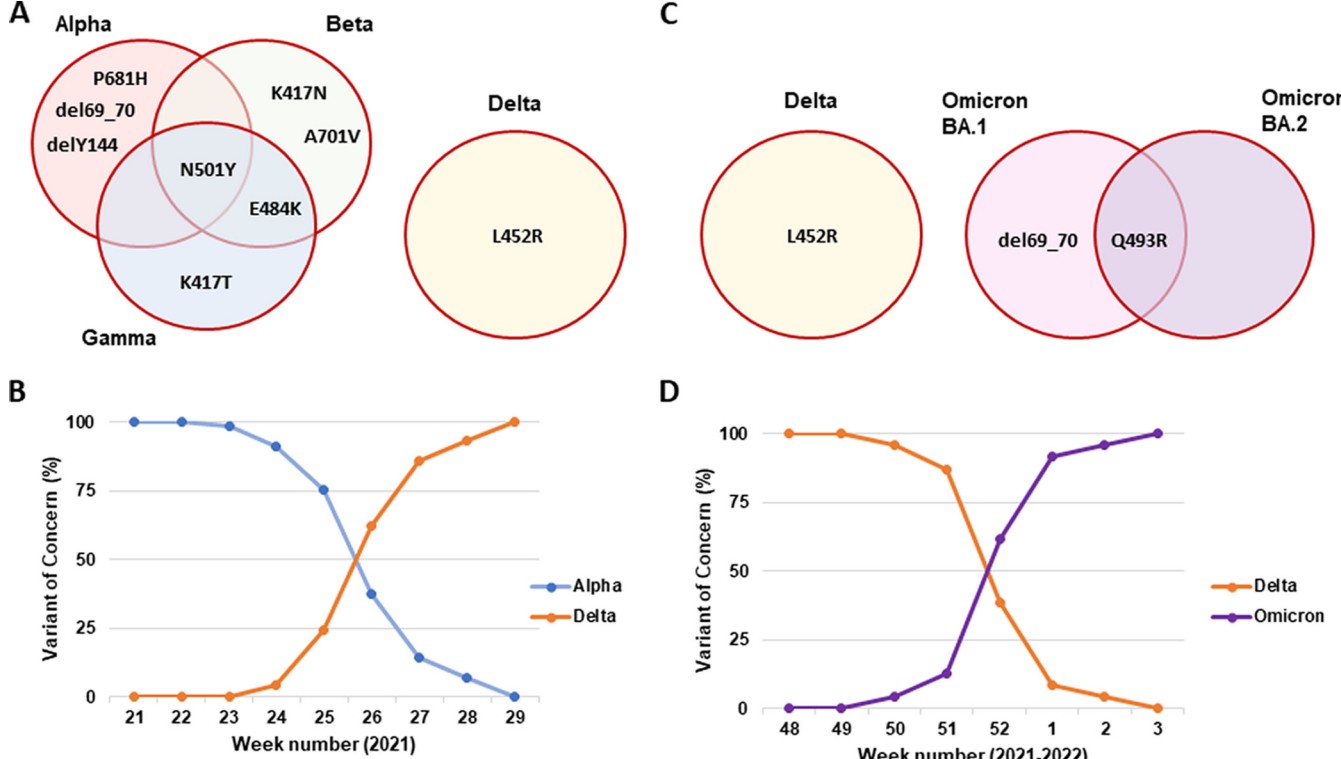

**FIG 1** SARS-CoV-2 variants of concern detected in The Netherlands, 2021-2022. (A) Nine RT-PCR genotyping assays that were selected to test the randomly selected SARS-CoV-2 positive samples in the period of May-July 2021 (N = 331). (B) Between May and July 2021 (N = 331), the Delta variant (N = 121) became dominant over the Alpha variant (N = 207). (C) Three RT-PCR genotyping assays that were selected to test the randomly selected SARS-CoV-2 positive samples in the period of December 2021-January 2022 (N = 333). (D) During December 2021 and January 2022 (N = 333), the Omicron variant (N = 129) became dominant over the Delta variant (N = 204).

**Accuracy of the RT-PCR genotyping assays for identification of Alpha, Beta, Gamma, and Delta SARS-CoV-2 VOC in comparison to WGS.** Taking into account the VOCs circulating at the time, we selected 9 RT-PCR genotyping assays to test 331 randomly selected SARS-CoV-2 positive samples collected in the period of May-July 2021 in 2 regions of The Netherlands, Flevoland and Limburg (Fig. 1A). Compared with the mutation profile, the RT-PCR genotyping assays classified 207/331 (62.5%) of the samples as Alpha, 121/331 (36.6%) as Delta, and 2/331 (0.6%) as Beta SARS-CoV-2 variants. One sample was classified as a non-VOC based on our selected RT-PCR genotyping assays. For all 331 samples, matching results were obtained using WGS (100%). The sample that could not be assigned to a specific lineage using RT-PCR genotyping assays was determined by WGS as the C.36.3.1 variant. Our data demonstrates the accuracy of the RT-PCR genotyping assays in VOC detection. Furthermore, our data set showed a rapid increase in Delta variant prevalence from 0% for the first week of June 2021 to 100% of SARS-CoV-2 cases by mid July 2021 (Fig. 1B).

**Tracking the increase in prevalence of the Omicron SARS-CoV-2 variant using RT-PCR genotyping assays in comparison to WGS.** After the first reports of Omicron SARS-CoV-2 variant, we included a novel RT-PCR genotyping assay for Omicron-specific mutation Q493R in our mutation panel to detect the Omicron SARS-CoV-2 positive samples, both subtypes BA.1 and BA.2, and differentiate them from the Delta variant which still accounted for almost all cases detected at the beginning of December 2021. Based on the VOCs circulating at the time, we selected 3 RT-PCR genotyping assays (Fig. 1C) and tested 333 SARS-CoV-2 positive samples collected between December 2021 and January 2022 in 4 Dutch regions of The Netherlands, Flevoland, Gelderland, Limburg, and Noord-Brabant. Based on the mutation profile, the RT-PCR genotyping assays classified 204/333 (61.3%) as Delta and 129/333 (38.7%) as Omicron SARS-CoV-2 variants, with 2 samples belonging to the BA.2 sub lineage of Omicron. In all 333 samples, matching

results were obtained using WGS (100%). These results demonstrated the accuracy of the RT-PCR genotyping assays in VOC detection. Furthermore, our data showed a shift of the Omicron variant prevalence from 0% to 100% in a time frame of 7 weeks (Fig. 1D).

**Phylogeny of collected sequences of SARS-CoV-2.** The phylogenetic tree containing 660 (near) full genome sequences and nine reference sequences shows clusters matching the 9 known SARS-CoV-2 clades included in this study: 19A (original Wuhan strain), 20D (non-VOC/VOI SARS-CoV-2 lineage C.36.3.1), 20H (Beta), 20I (Alpha), 20J (Gamma), 21I (Delta), 21J (Delta), 21K (Omicron, subtype BA.1), and 21L (Omicron, subtype BA.2). The concordance of the results obtained by WGS and RT-PCR genotyping was 100% (Fig. 2).

**Sensitivity of the RT-PCR genotyping assays versus WGS in SARS-CoV-2 samples with low viral loads.** To compare the sensitivity between both surveillance methods, RT-PCR genotyping assays and WGS were performed on 32 SARS-CoV-2 positive samples with high $C_T$ values ranging from 32 to 39 cycles. The RT-PCR genotyping assays were able to successfully classify SARS-CoV-2 VOCs in 30 out of 32 samples (93.8%), while WGS established the VOCs in 29 out of 32 samples (90.6%). One out of 32 samples failed both approaches (3.1%), whereas 3 samples failed 1 out of 2 methods (WGS:RT-PCR genotyping assays 2:1) (9.4%). However, the obtained results of those 3 samples were of poor quality, resulting in many and large regions with inconclusive sequences for WGS and the lack of a genotype call for the Delta-specific RT-PCR genotyping assay (L452R). Obtained results indicate similar sensitivity of RT-PCR genotyping assays to WGS for low viral load samples. For $C_{T-}$ values up to 35, both approaches were able to successfully classify the SARS-CoV-2 VOC.

## DISCUSSION

RT-PCR assays for the genotyping of viruses, such as hepatitis viruses (17), noroviruses (18), polioviruses (19), and rotaviruses (20), are widely used and the value of these assays has previously been shown. During the ongoing COVID-19 pandemic, genomic surveillance efforts have been performed worldwide and constitute an extremely valuable tool for monitoring SARS-CoV-2 evolution and its associated disease severity, as well as the performance of vaccines, therapeutics, diagnostic tools, or other public health measures. Although WGS is a critical tool for identification of new emerging variants, there is a need to develop faster, accurate, and more cost-effective methods for SARS-CoV-2 variant surveillance. This will enable quick public health responses and will also be applicable in lower resource settings.

This study evaluated the performance of a custom TaqMan SARS-CoV-2 mutation panel consisting of 10 RT-PCR genotyping assays detecting MOC based on allelic discrimination for SARS-CoV-2 VOC detection in comparison to WGS. The concordance between the 2 methods was 100%. Among 664 SARS-CoV-2 positive samples ($15 \leq C_T \leq 32$), both methods classified 31.2% as Alpha (N = 207); 0.3% as Beta (N = 2); 48.9% as Delta (N = 325), and 19.4% as Omicron (N = 129). Two out of 129 samples belonged to the BA.2 sub lineage of Omicron. One sample could not be assigned to a specific lineage and was classified as a non-VOC using RT-PCR genotyping assays. This sample was determined by WGS as the C.36.3.1 variant. Our data demonstrated the accuracy of genotyping assays for VOC surveillance. Moreover, RT-PCR genotyping assays were able to identify the shift of the Delta variant prevalence from 0% to 100% in a time frame of 8 weeks (May-July 2021) and a shift of the Omicron variant prevalence from 0% to 100% between December 2021 and January 2022 in The Netherlands, which further emphasized the relevance of genotyping assays for epidemiological surveillance.

Although our study provided SARS-CoV-2 VOC classification in a highly accurate manner, several limitations apply. First, genotyping assays rely on preexisting knowledge of the SARS-CoV-2 genomic sequences. Given the rapid emergence of new mutations and therefore new variants that can contain multiple co-occurring mutations, the design and selection of probe sets for genotyping assays will always depend on WGS.

While WGS is advocated by the WHO for the identification and confirmation of SARS-CoV-2 variants, this approach has several limitations as well. WGS has higher cost, longer

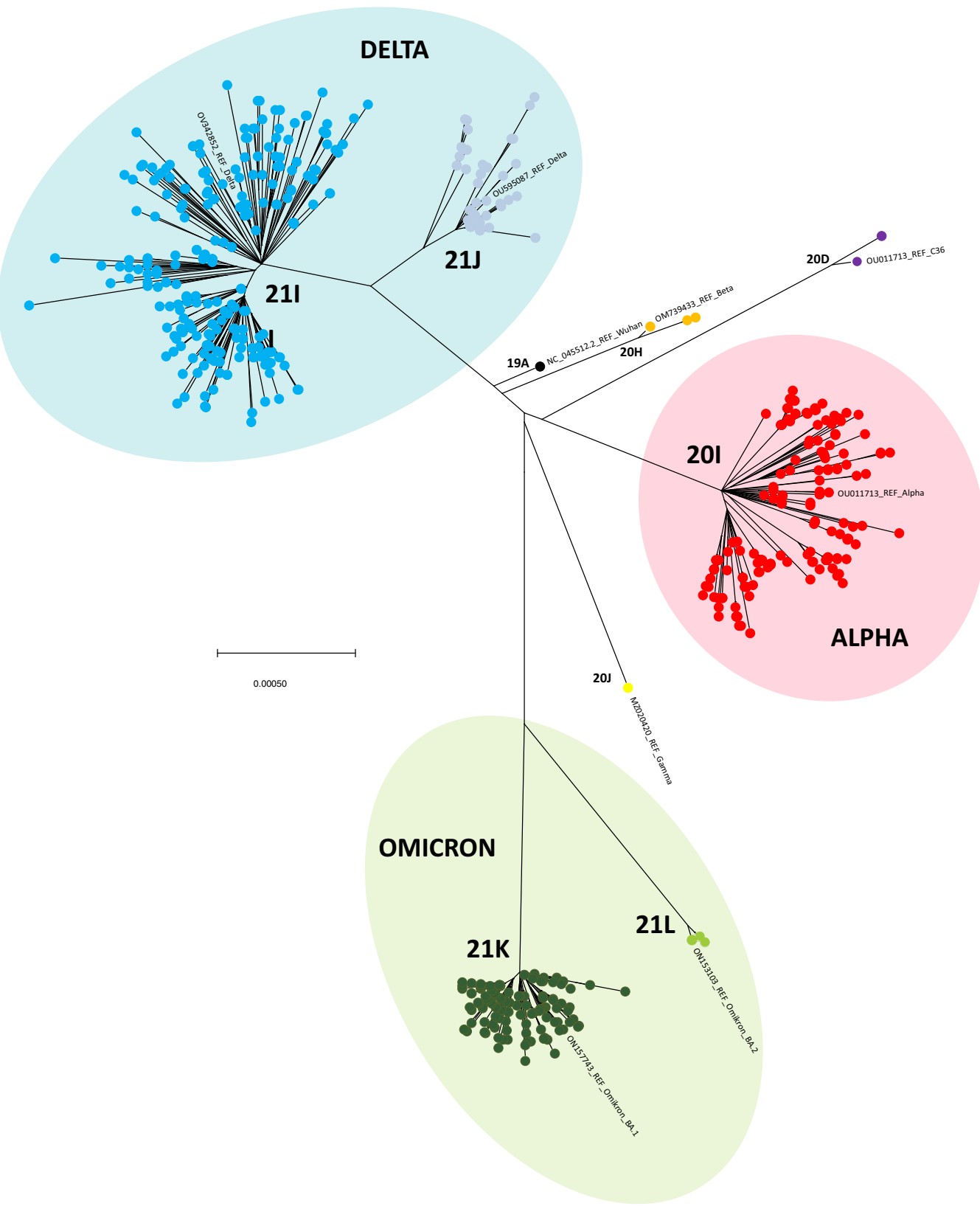

**FIG 2** Phylogenetic tree of SARS-CoV-2 variants of concern detected in The Netherlands, 2021-2022. A total of 669 SARS-CoV-2 genomes, including 660 of the 664 sequenced genomes from this study plus 9 reference sequences from GenBank, are depicted. Four genomes were excluded due to multiple regions with inconclusive sequencing. The sequences (N = 669) show clusters matching the 9 known SARS-CoV-2 clades included in this study: 19A (original Wuhan strain), 20D (non-VOC/VOI SARS-CoV-2 lineage C.36.3.1), 20H (Beta), 20I (Alpha), 20J (Gamma), 21I (Delta), 21J (Delta), 21K (Omicron, subtype BA.1), and 21L (Omicron, subtype BA.2).

turnaround times, and complexity of implementation in laboratories compared with RT-PCR genotyping assays. Thus, RT-PCR genotyping assays could constitute an alternative to WGS that allows a rapid turnaround time (<8 h versus 3 to 4 days by WGS), is easy to implement and cost-effective. The cost for the mutation-based approach even with a panel of 10 RT-PCR genotyping assays would be 15 times cheaper compared to WGS with possibility of greater savings if using a smaller panel. Once new mutations have been classified by WGS, custom RT-PCR genotyping assays could be set up within 2 weeks.

Another technique that could be considered for the routine detection of emerging SARS-CoV-2 VOC is a Multiplex PCR using melting curve analysis. Multiplex PCR is a useful method for the detection of SARS-CoV-2 VOC that produces reliable results but requires a nested PCR step which can increase the risk of contamination. In addition, allele-specific probes are used to perform multiplex PCR, that is why melting curves must be performed for confirmation (21). Furthermore, this technique may increase the risk of unclear or erroneous measurement due to the additional genetic variation in target regions which would require additional training of laboratory employees to interpret complex genetic information about a rapidly mutating virus (22, 23). As a result, genotyping assays could be easier to implement in laboratories than any currently available alternative methods.

Although the analytical sensitivity of the RT-PCR genotyping assays has not been measured in this study, Peterson et al. (2022) used a similar method to classify SARS-CoV-2 genomic variants in wastewater and they showed that the approach exhibited a significant sensitivity with the limits of detection (LOD) ranging from 3 to 6 copies/reaction (24). Similar studies using 1 or more genotyping assays to detect mutations in the *S* and/or *ORF8* gene reported the successful use of this approach for detection of SARS-CoV-2 VOC (25–27). Therefore, RT-PCR genotyping assays should be considered as an accurate and sensitive approach that could be executed in any standard laboratory.

Although PCR-based genotyping approaches will not replace WGS as the reference standard for the SARS-CoV-2 surveillance testing, they constitute an easily implementable and scalable tool complementary to WGS. Furthermore, the costs and turnaround time of genotyping approach are significantly reduced, and a new genotyping assay can be set up within 2 weeks when a new mutation is discovered by WGS. Finally, since evolution of viral mutation patterns in a country or region may indicate the emergence of new SARS-CoV-2 variants, conducting mutation surveillance using genotyping assays might be useful to identify a signal to start large scale WGS in a timely manner.

## MATERIALS AND METHODS

**Clinical nasopharyngeal samples and SARS-CoV-2 detection.** Nasopharyngeal swabs were collected daily from May to July 2021 and from December 2021 to January 2022 at the public health service centers in four Dutch provinces, Flevoland, Gelderland, Limburg, and Noord-Brabant. Upon sample collection, the swabs were placed into tubes containing 2 mL of lysis/binding buffer (Roche Diagnostic). The tubes were transported to the National Screening Laboratory of Sanquin (NSS) in Amsterdam, where the samples were tested for the presence of SARS-CoV-2 RNA using the cobas SARS-CoV-2 PCR test on the cobas 8800 System (Roche Diagnostics). The cobas SARS-CoV-2 PCR test amplifies 2 targets in *ORF1a/b* and *E* gene within the SARS-CoV-2 genome and contains internal control RNA molecules to check for nucleic acid extraction and amplification efficiency.

From all cobas SARS-CoV-2 PCR positive tested samples, 696 were randomly selected for inclusion in the study: N = 664 ($15 \leq C_T \leq 32$) to test the accuracy and N = 32 ($32 \leq C_T \leq 39$) to test the sensitivity of the RT-PCR genotyping assays.

To validate the RT-PCR genotyping assays (Thermo Fisher Scientific), a panel of SARS-CoV-2 positive samples (N = 13) was provided by The National Institute for Public Health and Environment (RIVM), covering the VOC Alpha, Beta, Gamma, Delta, and Omicron, former VOI Epsilon, Eta, Iota, and Zeta, and including the lineages B.1.177 (a major mostly European lineage, February 2020) and B.1.258.21 (The Netherlands lineage, October 2020) (https://cov-lineages.org/lineage_list.html).

The samples used in this study were obtained from individuals who visited public health service centers that were part of the national SARS-CoV-2 screening and surveillance system organized by the RIVM. Upon testing, visitors were informed that, if tested they positive for SARS-CoV-2, their samples could be used for virus typing to study the spread of SARS-CoV-2 within The Netherlands without further consent. Samples were anonymized prior to investigation. All procedures were carried out in accordance with the ethical standards of the Helsinki Declaration and The Netherlands Code of Conduct for Research Integrity.

**RNA extraction.** Regarding samples collected from May to July 2021, total RNA was extracted using the EasyMag total nucleic acid extractor (bioMérieux) according to the manufacturer's instructions. Briefly, 350 $\mu$L of the nasopharyngeal sample diluted in lysis/binding buffer was added to EasyMag

**TABLE 1** The 10 SARS-CoV-2 RT-PCR genotyping assays, detecting mutations of concern based on allelic discrimination of a custom TaqMan SARS-CoV-2 mutation panel which have been used in this study[a]

| | Spike protein mutations | | | | | | | | | |
|---|---|---|---|---|---|---|---|---|---|---|
| | N501Y | E484K | K417N | K417T | del69_70 | P681H | delY144 | A701V | L452R | Q493R |
| SARS-CoV-2 Variants of Concern | | | | | | | | | | |
| Alpha (B.1.1.7) | mut[b] | wt[c] | wt | wt | mut | mut | mut | wt | wt | wt |
| Beta (B.1.351) | mut | mut | mut | [d] | wt | wt | wt | mut | wt | wt |
| Gamma (P.1) | mut | mut | [d] | mut | wt | wt | wt | wt | wt | wt |
| Delta (B.1.617.2) | wt | wt | wt | wt | wt | [d] | wt | wt | mut | wt |
| Omicron (B.1.1.529) | | | | | | | | | | |
| BA.1 | mut | [d] | mut | [d] | mut | mut | [e] | wt | wt | mut |
| BA.2 | mut | [d] | mut | [d] | wt | mut | [e] | wt | wt | mut |

[a]The VOC lineage was determined based on the detected mutation profile.
[b]mut: mutation.
[c]wt: wildtype.
[d]The presence of another mutation at the same position would cause the assay to show a result that can look like a heterozygous call and can therefore signal the presence of a different mutation at that position.
[e]The delY144 assay was designed to detect the corresponding mutation in the Alpha variant of concern and could not be applied for the Omicron variant due to additional mutations being present in the adjacent positions that would compromise the ability of probes and primers to bind appropriately.

vessels containing 2 mL bioMérieux lysis buffer. After the addition of 50 $\mu$L silica, the samples were incubated at room temperature for 10 min. The elution volume was 55 $\mu$L. Nucleic acid extracts were stored at −80°C until further processing.

Samples collected between December 2021 and January 2022 underwent RNA extraction using the QIAsymphony DSP Virus/Pathogen Kit on a QiaSymphony instrument (Qiagen) according to the manufacturer's instructions. Samples were processed using the Complex400_V4_DSP protocol. A total of 400 $\mu$L of lysate was used and the elution volume was 85 $\mu$L. Nucleic acid extracts were stored at –80°C until further processing.

**Whole genome sequencing.** Whole-genome SARS-CoV-2 sequencing was performed using the Ion AmpliSeq SARS-CoV-2 research panel on an Ion S5 system (Thermo Fisher Scientific) according to the manufacturer's instructions. Samples were sorted in groups of 8 with comparable $C_T$ values as determined by the cobas SARS-CoV-2 PCR test on the cobas 8800 System. After nucleic acid extraction on the EasyMag or QiaSymphony, reverse transcription was performed on 10.5 $\mu$L of the total extracted RNA using the SuperScript VILO cDNA Synthesis Kit (Thermo Fisher Scientific). Ion AmpliSeq SARS-CoV-2 panel libraries were prepared using the Ion AmpliSeq Chef Reagents DL8 Kit and the Ion Chef Instrument (Thermo Fisher Scientific). Libraries of either 16 or 24 samples were pooled and run on 530 Chips (Thermo Fisher Scientific). Data were analyzed using the SARS_CoV_2_lineageID plugin based on the Pangolin COVID-19 Lineage Assigner on the Torrent Server or by uploading fasta files directly to the Pangolin COVID-19 Lineage Assigner (https://cov-lineages.org/resources/pangolin.html). SARS-CoV-2 sequences were submitted to the GISAID database (Table S1) (https://www.epicov.org/epi3/frontend#483d2c).

A phylogenetic tree was constructed containing (near) full genome sequences (N = 660; 15 $\leq$ $C_T$ $\leq$ 32) plus 9 reference sequences from GenBank matching the 9 SARS-CoV-2 clades of the (near) full genome sequences (Table S2). Four sequences of B.1.1.7 (N = 1) and BA.1 (N = 3) have not been included in the phylogenetic analysis due to multiple regions with inconclusive sequencing. After creating an alignment in Mega v. 11.0.8, a maximum likelihood (ML) approach with a general time reversed (GTR) model was used to construct the phylogeny. Bootstrap values (N = 100) were calculated to analyze the stability of the tree topology.

**RT-PCR genotyping assays.** Ten RT-PCR genotyping assays of the TaqMan SARS-CoV-2 mutation panel (Thermo Fisher Scientific) were selected as shown in Table 1. Assays were performed in a reaction volume of 10 $\mu$L using a QuantStudio 5 Real-time PCR instrument (QS5) (Applied Biosystems) following the manufacturer's protocol. For each sample, 5 $\mu$L of isolated SARS-CoV-2 RNA sample was added to 5 $\mu$L of TaqPath 1-step RT-PCR Master Mix containing 2 sequence-specific unlabeled primers and 2 allele-specific TaqMan probes. One TaqMan probe was specific for the wild type sequence and was VIC labeled, while the other probe was mutant-specific and labeled with FAM dye on the 5' end. The genotyping assays were run on the QS5 instrument using the following protocol: 50°C for 10 min, 95°C for 2 min, 45 cycles at 95°C for 3 s, and 60°C for 30 s. The data were analyzed using Design and Analysis Software 2.5.0 and the VOC lineage was determined based on the detected mutation profile (Table 1).

## SUPPLEMENTAL MATERIAL

Supplemental material is available online only.
**SUPPLEMENTAL FILE 1**, PDF file, 0.3 MB.

## ACKNOWLEDGMENTS

We thank Margret Sjerps and Ismail Maada (Sanquin Diagnostics) for their excellent technical assistance. We also thank Erik H. van Beers (Sanquin Diagnostics) for critical reading of the manuscript.

J.D.M.F., M.G., and O.S. are employees of Thermo Fisher Scientific and receive stock or equity interests. Thermo Fisher Scientific provided TaqMan SARS-CoV-2 mutation panel assays for the purpose of the study. J.D.M.F., M.G., and O.S. also provided technical expertise and assisted with data analysis and manuscript preparation and writing. The remaining authors declare that the research was conducted in the absence of any commercial or financial relationships that could be construed as a potential conflict of interest.

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
