## [Reviewer comments · Microbiology Spectrum]

Microbiology Spectrum

Efficient SARS-CoV-2 Surveillance during the Pandemic-Endemic Transition using PCR-based Genotyping Assays.

Lianne Koets, Karin van Leeuwen, Maaïke Derlagen, Jalenka van Wijk, Nadia Keijzer, Jelena Feenstra, Manoj Gandhi, Oceane Sorel, Thijs van de laar, and Marco Koppelman

Corresponding Author(s): Lianne Koets, Sanquin Bloedvoorziening

Review Timeline:

Submission Date:	August 29, 2022
Editorial Decision:	January 25, 2023
Revision Received:	February 16, 2023
Editorial Decision:	March 12, 2023
Revision Received:	March 14, 2023
Accepted:	April 14, 2023

Editor: Clinton Jones

Reviewer(s): Disclosure of reviewer identity is with reference to reviewer comments included in decision letter(s). The following individuals involved in review of your submission have agreed to reveal their identity: Ghulam Abbas (Reviewer #3); Jonathan Daniel Hulse (Reviewer #4)

Transaction Report:

DOI: <https://doi.org/10.1128/spectrum.03450-22>

January 25, 2023

Mx. Lianne Koets
Sanquin
National Screenings laboratory of Sanquin
Plesmanlaan 125
Amsterdam, Noord-Holland 1066 CX
Netherlands

Re: Spectrum03450-22 (Efficient SARS-CoV-2 Surveillance during the Pandemic-Endemic Transition using PCR-based Genotyping Assays.)

Dear Mx. Lianne Koets:

Thank you for submitting your manuscript to Microbiology Spectrum. As you will see your paper is very close to acceptance. Please modify the manuscript along the lines I have recommended. As these revisions are quite minor, I expect that you should be able to turn in the revised paper in less than 30 days, if not sooner. If your manuscript was reviewed, you will find the reviewers' comments below.

When submitting the revised version of your paper, please provide (1) point-by-point responses to the issues raised by the reviewers as file type "Response to Reviewers," not in your cover letter, and (2) a PDF file that indicates the changes from the original submission (by highlighting or underlining the changes) as file type "Marked Up Manuscript - For Review Only". Please use this link to submit your revised manuscript. Detailed instructions on submitting your revised paper are below.

Link Not Available

Sincerely,

Clinton Jones

Reviewer comments:

Reviewer #2 (Comments for the Author):

This manuscript entitled 'Efficient SARS-CoV-2 Surveillance during the Pandemic-Endemic Transition using PCR-based Genotyping Assays' is important and timely. The authors assessed the performance of a custom TaqMan SARS-CoV-2 mutation panel consisting of ten RT-PCR genotyping assays compared to whole genome sequencing (WGS) for identification of 5 VOC circulating in The Netherlands.

Among 664 SARS-CoV-2 positive samples, the RT-PCR genotyping assays classified 31.2% as Alpha (N=207); 48.9% as Delta (N=325); 19.4% as Omicron (N=129), 0.3% as Beta (N=2) and one sample as a non-VOC. Matching results were obtained using WGS in 100% of the samples.

The manuscript is well written, only a few modifications are suggested and highlighted in the word doc attached.

Reviewer #3 (Comments for the Author):

This is an important research and met analysis of Efficient SARS-CoV-2 Surveillance during the Pandemic-Endemic Transition using PCR-based Genotyping Assays, it pulls the data together in a way that is likely to be highly impactful and provides a state-of-the-art overview of the current knowledge of detecting COVID-19. It carries key information about the use of functions to design and monitor RT-PCR genotyping assays that would be a powerful method to include in SARS-CoV-2 surveillance testing and is likely to be highly cited in the future. However, the manuscript needs to be improved for certain necessary changes and spelling/grammar mistakes throughout and I support its publication after some minor changes. Moreover, I also would like to advise the author to strictly follow the journal's guidelines such as reference style etc. Moreover, I also request to add some references of work done on COVID-19 in Pakistan.

1-Give full names of all the abbreviations used first, then you can use abbreviations.

The English language of the article also needs to be improved

Italicize all the "et al., or et al." and scientific words (line 105).

The discussion part needs a more detailed analysis of the results.

Abbas G, Asif Iqbal 2*, Muhammad Arshad Javid 3, Waqar Saleem 2 and Muhammad Khurram Shahza. 2019. Covid-19 Attack, Prevention, Precaution and Managemental Strategies International Journal of Innovation and Research in Educational Sciences Volume 7, Issue 3, ISSN (Online): 2349-5219.

Ghulam Abbas ; Duraid K.A. AL-Taey ; Saad S.M.AL-Azawi ; Mohammad Mehdizadeh; Razia A. M. Qureshi; Ammar K Jasman; Ali K Slomy ; Mumtaz A. Khan ; Makhdoom Abdul Jabbar; Asif Iqbal; Maria Arshad; Jalees Ur Rehman1; Yusuf Konca ; Muhammad Arshad; Mehmood Ahmad. 2021. Controlling Strategies of Citrus to increase The Yield in the country: A step towards the fight against COVID-19. The paper is accepted to be published in " IOP Conference Series: Earth and Environmental Science" , Scopus index , cite score: 0.4

Ghulam, Abbas1; Robert, Odey Simon3; Razia AbdulMajid Qureshi6, Asif Iqbal2, Noor Fatima7; Emuru, Edward Odey5; Besong, Eric Ndoma. Beyond the Death and Infection Curriculum Vitae: DR GHULAM ABBAS Plagues of COVID-19 on the Globe: A Critical Analysis of its Diverse Effects. Manuscript number vmid-21-9686 accepted for publication in Virology and Mycology. Virol Mycol. Aff. 10: p768.

Muneer, M. A.; Munir, K.; Abbas, G.; Munir, I.; Khan, M. A.; Iqbal, A.; Ahmad, M. U. D.; Javid, M. A.; Fatima, Z.; Arshad, M. 2021. Facts and Figures on Covid-19 Pandemic Outbreak. Pakistan Journal of Zoology ; 53(3):1119-1129..

S. Areej1 , M. Ahmad*1, 2 , G. Abbas3 , A. Majeed1 , B. M. Beg1 , A. Iqbal4 , A. Basharat1, 2, R. M. Z. Mushtaq1,7 , W. Ahmad, S. Aroosa1 , S. Jaffery5 and S. B. Shabbir. 2022. A REVIEW ON ASPECTS OF CURRENT PHARMACOTHERAPIES FOR COVID-19. Pakistan Journal of Science. 74 (3):223-237.

Preparing Revision Guidelines

Please return the manuscript within 60 days; if you cannot complete the modification within this time period, please contact me. If you do not wish to modify the manuscript and prefer to submit it to another journal, please notify me of your decision immediately so that the manuscript may be formally withdrawn from consideration by Microbiology Spectrum.

Corresponding authors may join or renew ASM membership to obtain discounts on publication fees. Need to upgrade your

membership level? Please contact Customer Service at Service@asmusa.org.

Efficient SARS-CoV-2 Surveillance during the Pandemic-Endemic Transition using PCR-based Genotyping Assays

Lianne Koets¹, Karin van Leeuwen², Maaïke Derlagen³, Jalenka van Wijk³, Nadia Keijzer³, Jelena D. M. Feenstra⁴, Manoj Gandhi⁴, Oceane Sorel⁴, Thijs J. W. van de Laar^{5,6}, Marco Koppelman¹

¹Sanquin Research and Lab services, National Screening laboratory of Sanquin, 1066 CX, Amsterdam, The Netherlands

²Sanquin Diagnostics, Department of Phagocytes Diagnostics, 1066 CX, Amsterdam, The Netherlands

³Sanquin Diagnostics, Department of Immune Cytology, 1066 CX, Amsterdam, The Netherlands

⁴Thermo Fisher Scientific, South San Francisco, CA, 94080, USA

⁵Sanquin Research, Department of Donor Medicine Research, Laboratory of Blood Borne Infections, 1066 CX, Amsterdam, the Netherlands

⁶Onze Lieve Vrouwe Gasthuis, Laboratory of Medical Microbiology, 1091 AC, Amsterdam, The Netherlands

Corresponding author:

Lianne Koets

[revised manuscript text omitted]

values ranging from $32 \leq Ct \leq 39$ to test the sensitivity of these genotyping assays.

To validate the RT-PCR genotyping assays (Thermo Fisher Scientific, Waltham, Massachusetts, USA),
a panel of SARS-CoV-2 positive samples (N=13) was provided by The National Institute for Public Health and
Environment (RIVM), covering the VOC Alpha, Beta, Gamma, Delta and Omicron, former VOI Epsilon, Eta, Iota
and Zeta, and including the lineages B.1.177 (a major mostly European lineage, February 2020) and B.1.258.21
(The Netherlands lineage, October 2020) (https://cov-lineages.org/lineage_list.html).

The samples used in this study were obtained from individuals who visited public health service centres
that were part of the national SARS-CoV-2 screening and surveillance system organised by the RIVM. Upon
testing, visitors were informed that, if tested SARS-CoV-2 positive, their samples could be used for virus typing
to study the spread of SARS-CoV-2 within the Netherlands without further consent. Samples were anonymised
prior to investigation. All procedures were carried out in accordance with the ethical standards of the Helsinki
Declaration and the Netherlands Code of Conduct for Research Integrity.

**RNA extraction**

Regarding samples collected from May to July 2021, total RNA was extracted using the EasyMag total nucleic
acid extractor (bioMérieux, Marcy l'Etoile, France) according to the manufacturer's instructions. Briefly, 350 µL
of the nasopharyngeal sample diluted in lysis/binding buffer was added to EasyMag vessels containing 2 mL
bioMerieux lysis buffer. After addition of 50 µL silica, the samples were incubated at room temperature for 10
minutes. Next, samples were processed according to the **specific B2.0.1. protocol**. The elution volume was 55
120 µL. Nucleic acid extracts were stored at **<-80°C** until further processing.

Samples collected between December 2021 and January 2022 underwent RNA extraction using the
QIASymphony DSP Virus/Pathogen kit on a QiaSymphony instrument (all Qiagen, Hilden, Germany) according
to the manufacturer's instructions. Samples were processed using the Complex400_V4_DSP protocol. 400 µL
of lysate was used and the elution volume was 85 µL. Nucleic acid extracts were stored at <-80°C until further
processing.

**Whole genome sequencing**

Whole genome SARS-CoV-2 sequencing was performed using the Ion AmpliSeq SARS-CoV-2 research panel
on an Ion S5 system (Thermo Fisher Scientific, Waltham, Massachusetts, USA) according to the manufacturer's
instructions. Samples were sorted in groups of eight with comparable Ct-values as determined by the cobas
SARS-CoV-2 PCR test on the cobas 8800 System. After nucleic acid extraction on the EasyMag or
QiaSymphony, reverse transcription was performed on 10.5 µL of the total extracted RNA using the SuperScript
VILO cDNA Synthesis Kit. Ion AmpliSeq SARS-CoV-2 panel libraries were prepared using the Ion AmpliSeq
Chef Reagents DL8 Kit and the Ion Chef Instrument (**all** Thermo Fisher Scientific, Waltham, Massachusetts,
USA). Libraries of either 16 or 24 samples were pooled and run on 530 Chips (Thermo Fisher Scientific,
Waltham, Massachusetts, USA). Data were analyzed using the SARS_CoV_2_lineageID plugin based on the
Pangolin COVID-19 Lineage Assigner on the Torrent Server or by uploading fasta files directly to the Pangolin
COVID-19 Lineage Assigner (<https://cov-lineages.org/resources/pangolin.html>). SARS-CoV-2 sequences were
submitted to the GISAID database (Supplementary Table S1) (<https://www.epicov.org/epi3/frontend#483d2c>).

A phylogenetic tree was constructed containing (near) full genome sequences (N=660; 15≤Ct≤32) plus
nine reference sequences from GenBank matching the nine SARS-CoV-2 clades of the (near) full genome
sequences (Supplementary Table S2). Four sequences of B.1.1.7 (N=1) and BA.1 (N=3) had not been included
in the phylogenetic analysis due to multiple regions with inconclusive sequencing (a high number of N's). After

creating an alignment in Mega v. 11.0.8, a maximum likelihood (ML) approach with a general time reversed
(GTR) model was used to construct the phylogeny. Bootstrap values (N=100) were calculated to analyze the
stability of the tree topology.

**RT-PCR genotyping assays**

Ten RT-PCR genotyping assays of the TaqMan SARS-CoV-2 mutation panel (Thermo Fisher Scientific,
Waltham, Massachusetts, USA) were selected as shown in Table 1. Assays were performed in a reaction
volume of 10 μ L using a QuantStudio 5 Real-time PCR instrument (QS5) (Applied Biosystems, CA, USA)
following the manufacturer's protocol. For each sample, 5 μ L of isolated SARS-CoV-2 RNA sample was added
to 5 μ L of TaqPath 1-step RT-PCR Master Mix containing two sequence-specific unlabeled primers and two
allele-specific TaqMan probes. One TaqMan probe is specific for the wild type sequence and is VIC labeled,

[revised manuscript text omitted]

Lianne Koets¹, Karin van Leeuwen², Maaïke Derlagen³, Jalenka van Wijk³, Nadia Keijzer³, Jelena D. M.
Feenstra⁴, Manoj Gandhi⁴, Oceane Sorel⁴, Thijs J. W. van de Laar^{5,6}, Marco Koppelman¹

¹Sanquin Research and Lab services, National Screening laboratory of Sanquin, 1066 CX, Amsterdam, The
Netherlands

²Sanquin Diagnostics, Department of Phagocytes Diagnostics, 1066 CX, Amsterdam, The Netherlands

³Sanquin Diagnostics, Department of Immune Cytology, 1066 CX, Amsterdam, The Netherlands

⁴Thermo Fisher Scientific, South San Francisco, CA, 94080, USA

⁵Sanquin Research, Department of Donor Medicine Research, Laboratory of Blood Borne Infections, 1066 CX,
Amsterdam, the Netherlands

⁶Onze Lieve Vrouwe Gasthuis, Laboratory of Medical Microbiology, 1091 AC, Amsterdam, The Netherlands

Corresponding author:

Lianne Koets

[revised manuscript text omitted]

values ranging from $32 \leq Ct \leq 39$ to test the sensitivity of these genotyping assays.

To validate the RT-PCR genotyping assays (Thermo Fisher Scientific, Waltham, Massachusetts, USA),
a panel of SARS-CoV-2 positive samples (N=13) was provided by The National Institute for Public Health and
Environment (RIVM), covering the VOC Alpha, Beta, Gamma, Delta and Omicron, former VOI Epsilon, Eta, Iota
and Zeta, and including the lineages B.1.177 (a major mostly European lineage, February 2020) and B.1.258.21
(The Netherlands lineage, October 2020) (https://cov-lineages.org/lineage_list.html).

The samples used in this study were obtained from individuals who visited public health service centers
that were part of the national SARS-CoV-2 screening and surveillance system organized by the RIVM. Upon
testing, visitors were informed that, if tested SARS-CoV-2 positive, their samples could be used for virus typing
to study the spread of SARS-CoV-2 within the Netherlands without further consent. Samples were anonymized
prior to the investigation. All procedures were carried out under the ethical standards of the Helsinki Declaration
and the Netherlands Code of Conduct for Research Integrity.

**RNA extraction**

Regarding samples collected from May to July 2021, total RNA was extracted using the EasyMag total nucleic
acid extractor (bioMérieux, Marcy l'Etoile, France) according to the manufacturer's instructions. Briefly, 350 µL
of the nasopharyngeal sample diluted in lysis/binding buffer was added to EasyMag vessels containing 2 mL
bioMerieux lysis buffer. After the addition of 50 µL silica, the samples were incubated at room temperature for
10 minutes. Next, samples were processed according to the specific B2.0.1. protocol. The elution volume was
55 µL. Nucleic acid extracts were stored at <-80°C until further processing.

Samples collected between December 2021 and January 2022 underwent RNA extraction using the
QIA Symphony DSP Virus/Pathogen kit on a QIA Symphony instrument (all Qiagen, Hilden, Germany) according
to the manufacturer's instructions. Samples were processed using the Complex400_V4_DSP protocol. 400 µL
of lysate was used and the elution volume was 85 µL. Nucleic acid extracts were stored at <-80°C until further
processing.

**Whole genome sequencing**

Whole genome SARS-CoV-2 sequencing was performed using the Ion AmpliSeq SARS-CoV-2 research panel
on an Ion S5 system (Thermo Fisher Scientific, Waltham, Massachusetts, USA) according to the manufacturer's
instructions. Samples were sorted in groups of eight with comparable Ct-values as determined by the cobas
SARS-CoV-2 PCR test on the cobas 8800 System. After nucleic acid extraction on the EasyMag or
QIA Symphony, reverse transcription was performed on 10.5 µL of the total extracted RNA using the SuperScript
VILO cDNA Synthesis Kit. Ion AmpliSeq SARS-CoV-2 panel libraries were prepared using the Ion AmpliSeq
Chef Reagents DL8 Kit and the Ion Chef Instrument (all Thermo Fisher Scientific, Waltham, Massachusetts,
USA). Libraries of either 16 or 24 samples were pooled and run on 530 Chips (Thermo Fisher Scientific,
Waltham, Massachusetts, USA). Data were analyzed using the SARS_CoV_2_lineageID plugin based on the
Pangolin COVID-19 Lineage Assigner on the Torrent Server or by uploading fasta files directly to the Pangolin
COVID-19 Lineage Assigner (<https://cov-lineages.org/resources/pangolin.html>). SARS-CoV-2 sequences were
submitted to the GISAID database (Supplementary Table S1) (<https://www.epicov.org/epi3/frontend#483d2c>).

A phylogenetic tree was constructed containing (near) full genome sequences (N=660; 15≤Ct≤32) plus
nine reference sequences from GenBank matching the nine SARS-CoV-2 clades of the (near) full genome
sequences (Supplementary Table S2). Four sequences of B.1.1.7 (N=1) and BA.1 (N=3) had not been included
in the phylogenetic analysis due to multiple regions with inconclusive sequencing (a high number of N's). After

creating an alignment in Mega v. 11.0.8, a maximum likelihood (ML) approach with a general time-reversed
(GTR) model was used to construct the phylogeny. Bootstrap values (N=100) were calculated to analyze the
stability of the tree topology.

**RT-PCR genotyping assays**

Ten RT-PCR genotyping assays of the TaqMan SARS-CoV-2 mutation panel (Thermo Fisher Scientific,
Waltham, Massachusetts, USA) were selected as shown in Table 1. Assays were performed in a reaction
volume of 10 μ L using a QuantStudio 5 Real-time PCR instrument (QS5) (Applied Biosystems, CA, USA)
following the manufacturer's protocol. For each sample, 5 μ L of isolated SARS-CoV-2 RNA sample was added
to 5 μ L of TaqPath 1-step RT-PCR Master Mix containing two sequence-specific unlabeled primers and two
allele-specific TaqMan probes. One TaqMan probe is specific for the wild-type sequence and is VIC labeled,
while the other probe is mutant-specific and labeled with FAM dye on the 5' end. The genotyping assays were
run with the QS5 instrument using the following protocol: 50°C for 10 min, 95°C for 2 min, 45 cycles at 95°C for
3 s and 60°C for 30 s. Post-readings were carried out at 50°C. The data were analyzed using Design and
Analysis Software 2.5.0 and the VOC lineage was determined based on the detected mutation profile (Table 1).

**Results**

**Validation of the RT-PCR genotyping assays using reference material**

To validate the accuracy of selected RT-PCR genotyping assays of the TaqMan SARS-CoV-2 mutation panel,
we tested a panel of reference samples (N=13) provided by the RIVM containing SARS-CoV-2 positive samples
with known whole genome sequences and therefore known types of selected SARS-CoV-2 variants: Alpha,
Beta, Gamma, Delta, Epsilon, Eta, Iota, Omicron, Zeta, as well as lineages B.1.258.21 and B.1.177. All samples
were tested using selected genotyping assays and the results showed 100% concordance (Supplementary
Table S3).

[revised manuscript text omitted]

		Spike protein mutations									
		N501Y	E484K	K417N	K417T	del169_70	P681H	delY144	A701V	L452R	Q493R
SARS-CoV-2 Variants of Concern	Alpha (B.1.1.7)	mut	wt	wt	wt	mut	mut	mut	wt	wt	wt
	Beta (B.1.351)	mut	mut	mut	*	wt	wt	wt	mut	wt	wt
	Gamma (P.1)	mut	mut	*	mut	wt	wt	wt	wt	wt	wt
	Delta (B.1.617.2)	wt	wt	wt	wt	wt	*	wt	wt	mut	wt
	Omicron (B.1.1.529)	BA.1	mut	*	mut	*	mut	mut	/	wt	wt
BA.2		mut	*	mut	*	wt	mut	/	wt	wt	mut

With mut: mutation, wt: wildtype, *: the presence of another mutation at the same position would cause the assay to
 show a result which can look as a heterozygous call and can therefore signal the presence of a different mutation at
 that position, /: the delY144 assay was designed to detect the corresponding mutation in the Alpha variant of concern
 and could not be applied for the Omicron variant due to additional mutations being present in the adjacent positions
 that would compromise the ability of probes and primers to bind appropriately.

**Footnotes**

All authors report no conflicts of interest related to this work.

This study was not funded.

Response to Reviewers

Reviewer #2

- Only a few modifications are suggested and highlighted in the word doc attached.

Reply: The suggested modifications/highlighted sentences have been corrected.

Reviewer #3

- Give full names of all the abbreviations used first, then you can use abbreviations.

Reply: All full names are given before using the abbreviations.

- The English language of the article also needs to be improved.

Reply: The manuscript has been carefully reviewed. Grammar and spelling errors have been corrected.

- Italicize all the "et al." and scientific words.

Reply: All scientific words and "et al." are now written in *Italic*.

- Follow the journal's guidelines such as reference style

Reply: The reference style has been changed to American Society for Microbiology (ASM)

- The discussion part needs a more detailed analysis of the results.

Reply: This was a very good point of reviewer 3. The analysis of the results has now been explained in more detail.

- I also request to add some references of work done on COVID-19 in Pakistan.

Reply: We read the suggested articles. However, we did not add them as a reference. The articles were very interesting, but too general to include in our manuscript.

March 12, 2023

Mx. Lianne Koets
Sanquin Bloedvoorziening
National Screening laboratory of Sanquin
Plesmanlaan 125
Amsterdam, Noord-Holland 1066 CX
Netherlands

Re: Spectrum03450-22R1 (Efficient SARS-CoV-2 Surveillance during the Pandemic-Endemic Transition using PCR-based Genotyping Assays.)

Dear Mx. Lianne Koets:

Thank you for submitting your manuscript to Microbiology Spectrum. As you will see your paper is very close to acceptance. Please modify the manuscript along the lines I have recommended. As these revisions are quite minor, I expect that you should be able to turn in the revised paper in less than 30 days, if not sooner. If your manuscript was reviewed, you will find the reviewers' comments below.

When submitting the revised version of your paper, please provide (1) point-by-point responses to the issues raised by the reviewers as file type "Response to Reviewers," not in your cover letter, and (2) a PDF file that indicates the changes from the original submission (by highlighting or underlining the changes) as file type "Marked Up Manuscript - For Review Only". Please use this link to submit your revised manuscript. Detailed instructions on submitting your revised paper are below.

Link Not Available

Sincerely,

Clinton Jones

Reviewer comments:

Reviewer #3 (Comments for the Author):

Thanks for incorporating the changes

Reviewer #4 (Comments for the Author):

Overall, this will add to our understanding of SARS-CoV - 2 detection. The paper is well written, and is very concise. There are a few small issues, that can easily be corrected.

Line 44 - 46: Run-on sentence, do not use 'and' multiple times in a sentence. Break these into multiple sentences.

Line 48-50: Run-on sentence, do not use 'and' multiple times in a sentence. Break these into multiple sentences.

Line 52: Tab over to start a new paragraph

Line 61-63: Run-on sentence, do not use 'and' multiple times in a sentence. Break these into multiple sentences.

Line 98- 100: Run-on sentence, do not use 'and' multiple times in a sentence. Break these into multiple sentences.

Line 212 - 215: Run-on sentence, do not use 'and' multiple times in a sentence. Break these into multiple sentences.

Line 215 - 218: Run-on sentence, do not use 'and' multiple times in a sentence. Break these into multiple sentences.

Preparing Revision Guidelines

Please return the manuscript within 60 days; if you cannot complete the modification within this time period, please contact me. If you do not wish to modify the manuscript and prefer to submit it to another journal, please notify me of your decision immediately so that the manuscript may be formally withdrawn from consideration by Microbiology Spectrum.

Efficient SARS-CoV-2 Surveillance during the Pandemic-Endemic Transition using PCR2 based Genotyping Assays

Line 44 – 46: Run-on sentence, do not use ‘and’ multiple times in a sentence. Break these into multiple sentences.

Line 48-50: Run-on sentence, do not use ‘and’ multiple times in a sentence. Break these into multiple sentences.

Line 52: Tab over to start a new paragraph

Line 61-63: Run-on sentence, do not use ‘and’ multiple times in a sentence. Break these into multiple sentences.

Line 98- 100: Run-on sentence, do not use ‘and’ multiple times in a sentence. Break these into multiple sentences.

Line 212 – 215: Run-on sentence, do not use ‘and’ multiple times in a sentence. Break these into multiple sentences.

Line 215 – 218: Run-on sentence, do not use ‘and’ multiple times in a sentence. Break these into multiple sentences.

Response to Reviewers

Reviewer #4

- Line 44 - 46: Run-on sentence, do not use 'and' multiple times in a sentence. Break these into multiple sentences.

Reply: The run-on sentence has been corrected.

- Line 48-50: Run-on sentence, do not use 'and' multiple times in a sentence. Break these into multiple sentences.

Reply: The run-on sentence has been corrected.

- Line 52: Tab over to start a new paragraph.

Reply: This has been corrected.

- Line 61-63: Run-on sentence, do not use 'and' multiple times in a sentence. Break these into multiple sentences.

Reply: The run-on sentence has been corrected.

- Line 98- 100: Run-on sentence, do not use 'and' multiple times in a sentence. Break these into multiple sentences.

Reply: The run-on sentence has been corrected.

- Line 212 - 218: Run-on sentences, do not use 'and' multiple times in a sentence. Break these into multiple sentences.

Reply: The run-on sentence has been corrected.

April 14, 2023

Mx. Lianne Koets
Sanquin Bloedvoorziening
National Screening laboratory of Sanquin
Plesmanlaan 125
Amsterdam, Noord-Holland 1066 CX
Netherlands

Re: Spectrum03450-22R2 (Efficient SARS-CoV-2 Surveillance during the Pandemic-Endemic Transition using PCR-based Genotyping Assays.)

Dear Mx. Lianne Koets:

Your manuscript has been accepted, and I am forwarding it to the ASM Journals Department for publication. You will be notified when your proofs are ready to be viewed.

Sincerely,

Clinton Jones
Editor, Microbiology Spectrum